# Destabilizing effects on a classic tri-trophic oyster-reef cascade

**Virginia R. Schweiss, Chet F. Rakocinski**⊙*

Division of Coastal Sciences, School of Ocean Science and Engineering, University of Southern Mississippi, Gulf Coast Research Laboratory, Ocean Springs, Mississippi, United States of America

* chet.rakocinski@usm.edu

**Data Availability Statement:** Data used for this study were obtained from experiments as described in the Methods and materials section of this article. All data were collected, compiled, curated and validated by V.R. Schweiss and C.F. Rakocinski. All data reported in this manuscript are

## Abstract

How interactions among multiple predators affect the stability of trophic cascades is a topic of special ecological interest. To examine factors affecting the stability of the classic tri-trophic oyster reef cascade within a different context, configurations of three predators, including the Gulf toadfish, Gulf stone crab, and oystershell mud crab, were manipulated together with either oyster shell or limestone gravel substrate within a multiple predator effects (MPE) experiment. Additionally, a complimentary set of trait-mediated-indirect interaction (TMII) experiments examined the inhibition of oyster consumption relative to mud-crab size and top predator identity in the absence of other cues and factors. The classic tri-trophic cascade formed by the toadfish-mud crab-oyster configuration was potentially weakened by several interactions within the MPE experiment. Consumption of oysters and mud crabs by the intra-guild stone crab was undeterred by the presence of toadfish. Although mud crab feeding was inhibited in the presence of both toadfish and stone crabs, estimated non-consumptive effects (NCEs) were weaker for stone crabs in the MPE experiment. Consequently, the total effect was destabilizing when all three predator species were together. Inhibition of mud crab feeding was inversely related to direct predation on mud crabs within the MPE experiment. Complimentary TMII experiments revealed greater inhibition of mud crab feeding in response to stone crabs under sparse conditions. TMII experiments also implied that inhibition of mud crab feeding could have largely accounted for NCEs relative to oysters within the MPE experiment, as opposed to interference by other mud crabs or top predators. An inverse relationship between mud crab size and NCE strength in the TMII experiment disclosed another potentially destabilizing influence on the tri-trophic-cascade. Finally, although habitat complexity generally dampened the consumption of oysters across MPE treatments, complex habitat promoted mud crab feeding in the presence of toadfish alone. This study underscores how ecological interactions can mediate trophic cascades and provides some additional insights into the trophic dynamics of oyster reefs for further testing under natural conditions.

## Introduction

Communities are structured through strong biotic interactions, including the top-down effects of predation [1, 2]. Top predators may stabilize basal prey populations (i.e., decrease chances

publicly available, including metadata formats, and are accessible as SPSS files at the Dryad Digital Repository, doi:10.5061/dryad.47d7wm3b5.

**Funding:** Research for this article was conducted as part of a Ph.D. Dissertation by VRF, and was not supported by any specific grant from funding agencies in the public, commercial, or not-for-profit sectors. Supplies for this research project were largely provided to VRF by the Mississippi Department of Marine Resources.

**Competing interests:** The authors have no competing interests.

of basal prey extirpation) by suppressing the interaction between intermediate predators and basal prey within simple tri-trophic cascades [3–6]. However, the stability of trophic cascades become less predictable as more interacting species are incorporated [7]. How the stability of trophic cascades is affected by interactions among multiple predator species is a topic of special ecological interest [8, 9].

Top predators indirectly enhance the abundances of basal prey within trophic cascades [10]. Basal prey organisms benefit indirectly from the consumption of intermediate predators by top predators through a process known as density-mediated indirect interactions (DMIIs) [11–14]. Moreover, basal prey organisms benefit indirectly from the suppression of feeding by intermediate predators in the presence of top predators through a process known as trait-mediated indirect interactions (TMIIs) [3, 5, 6]. Non-consumptive effects (NCEs) arise from non-lethal changes in the biology or behavior of intermediate predators that lead to their reduced fitness [13, 15, 16]. For example, reduced feeding due to predator avoidance reduces the energy available for growth and reproduction [14, 17].

Multiple Predator Effects (MPEs) arise from facilitative and competitive interactions among multiple predators that can have non-additive effects on basal prey abundance [18, 19]. Synergistic MPEs cannot be foreseen from the effects of single predators in isolation [14, 20]. Moreover, interactions among multiple predators may affect the stability of trophic cascades as a function of basal prey mortality [21]. Facilitation between top predators can destabilize trophic cascades by accelerating feeding on a basal resource [19]. Conversely, interference or predation between intraguild predators (i.e., species feeding on a common prey) [14, 22] or size classes can decelerate feeding on a common basal resource [23]. Therefore, to discern the combined effects on shared basal prey, multiple predators need to be studied jointly [9].

Trophic cascade stability potentially responds to any traits that mediate the risk of mortality to intermediate predators [24]. Body size is an overarching trait often overlooked within food web studies [25]. Inhibitive effects of top predators on feeding by intermediate predators should vary with body size because predation risk correlates directly with the predator-prey size ratio [22]. As such, TMII strength should vary with the size of intermediate predators [26]. Accordingly, predator interactions may induce changes in the size selection of basal prey [20]. Additionally, interference or predatory interactions among intraguild predators are strongly mediated by body size, which adds a horizontal dimension within food webs [23].

Habitat complexity also mediates trophic cascades by weakening or reinforcing predation and interference interactions [14, 27–29]. Structurally complex habitat reduces predator efficiency by providing refuge for basal prey, while also protecting intermediate predators [30–34]. Complex habitat often stabilizes trophic cascades by reducing predation on basal prey [17, 35, 36]. Conversely, complex habitat may destabilize trophic cascades by reducing interference between intraguild predators [22, 29, 36–38]. Thus, the degree to which the trophic cascade is stabilized by habitat complexity also depends on the identities and body sizes of intermediate and top predators [9].

The oyster reef complex offers an ideal model system for the study of trophic interactions in connection with effects on basal oyster prey [9, 14, 19, 22, 23]. Core assemblages comprising multiple species and sizes of top and intermediate predators can be readily manipulated [35]. Interference, predation, and facilitation interactions within assemblages can modulate abundances of basal oysters in unexpected ways [19]. Trophic interactions are also mediated by habitat complexity [14]. Indeed, previous studies of oyster reef trophic cascades reveal important context-dependence relative to predator identity [9, 19], predator size [22, 23], and habitat complexity [14, 22].

Experimental studies of oyster reefs typically consider subsets of predators forming tri-trophic cascades [sensu 17] or triangular food web compartments [sensu 19], within which

competitive, predatory, or facilitative interactions influence cascade stability [39]. A classic tri-trophic cascade within the oyster-reef complex comprises the oyster toadfish (*Opsanus tau*), the mud crab (*Panopeus* spp.), and the juvenile oyster (*Crassostrea virginica*). Oyster toadfish stabilize the trophic cascade by inhibiting mud crab feeding on juvenile oysters, although the efficacy of the cascade can be context-dependent [40]. Interactions involving other predator species can also affect the stability of the tri-trophic cascade [14].

A common approach to disentangling MPEs involves experimentally manipulating combinations of different types and sizes of predators [22, 23]. As such, the effects of complex interactions have not been examined for the multiple predator configuration involving the Gulf toadfish (*Opsanus beta*), the Gulf stone crab (*Menippe adina*), and the oystershell mud crab (*Panopeus simpsoni*). As top predators, toadfish consume mud crabs and juvenile stone crabs, but not oysters [41]. Stone crabs are intraguild predators [sensu 42] of mud crabs [9], and mud crabs potentially interfere with [14] and consume each other [23]. Both mud crabs and stone crabs share juvenile oysters as prey [43].

Previous studies illustrate how various ecological interactions affect the stability of the oyster reef tri-trophic cascade [9, 17, 19, 22, 23]. Here we focus on several effects on the oyster tri-trophic cascade within a different context involving the addition of an intraguild predator, the Gulf stone crab, two levels of habitat complexity characterized by oyster shell vs. limestone gravel substrate, and TMII effects of toadfish vs. stone crab relative to mud crab size. To examine these effects, two complementary laboratory experiments were conducted. Effects of predator identity configuration and habitat complexity were examined through mesocosm trials involving exclusive combinations of Gulf toadfish, Gulf stone crab, the oystershell mud crab, two size classes of juvenile oysters, and two substrate types (oyster shell vs. limestone gravel). Trait mediated indirect interaction (TMII) effects attributable strictly to the inhibition of mud crab feeding by top predators were examined through complimentary experiments involving caged toadfish or stone crabs as top predators and different sizes of mud crabs as intermediate predators.

## Materials and methods

### Sources of test animals

Animals for experiments were obtained using plastic sampling trays (52 cm x 52 cm x 11 cm) lined with self-closing mosquito netting (0.5 mm mesh) and filled with loose oyster shells ($\approx$ 6,000 g). Trays were deployed *in situ* for 10 weeks at twelve nearshore subtidal reef sites located along the entire Mississippi coast [44]. Sites were collocated as pairs of historically harvested and low-profile substrate-augmented reefs geographically dispersed within the western, central, and eastern regions of the Mississippi coast, falling between just west of Saint Louis Bay (30.30030˚ N, -89.32930˚ W) and near the mouth of the Pascagoula River (30.36453˚ N, -88.59988˚ W).

Live animals for experiments were identified, measured, and placed into holding tanks. Test animals were acclimated at 25˚C and 29 salinity for at least three days. Individual crabs were kept isolated to prevent aggressive interactions. Reared juvenile oysters for experiments were obtained from the University of Auburn Shellfish Laboratory in Dauphin Island, AL.

### MPE experiment

Oyster survival was examined as a response to different combinations of top and intermediate predators, and two types of substrate, oyster shell or #57 limestone gravel. The substrate was soaked and rinsed to remove organic debris before using it. Small mesocosms (89 x 48 x 33 cm tubs) filled with 110 l filtered seawater served as experimental units. One hour before the

introduction of test animals, 30 loose juvenile oysters (15 spat, 15 mm shell height (SH) and 15 seed, 30 mm SH) were randomly distributed upon the substrate. Heaters maintained the water temperature at 25˚C; aerators provided water circulation and oxygenation. Abiotic variables were monitored with a YSI model 85 handheld meter. Sizes of test animals were standardized: mud crabs ≈ 23 mm carapace width (CW), stone crabs ≈ 60 mm CW, and toadfish ≈ 120 mm total length. Top predators matched the most common sizes capable of consuming mud crabs. Oyster densities ensured total prey depletion did not occur within the 24 h trial duration. Four randomized blocks comprising all MPE treatment combinations were run separately for each substrate type: (1) 30 oyster (control); (2) seven mud crabs (MC); (3) seven MC and one toadfish (TF); (4) seven MC and one stone crab (SC); (5) one SC; (6) one SC and one TF; (7) seven MC, one TF, and one SC. The TF and oyster combination was not included, as toadfish do not consume oysters [40, 41]. Following the completion of trials, surviving oysters and mud crabs were enumerated. Test animals were not reused to ensure independence.

## TMII experiments

Non-consumptive effects (NCE) estimated from the MPE experiment (see below) could result from multiple factors including behavioral inhibition, interference, and interactions between the top and intermediate predators, as mediated by substrate complexity. Therefore, a complementary set of trait-mediated indirect interaction (TMII) experiments focused on responses by different sizes of intermediate mud crabs under sparse (i.e., lacking other mediating factors) conditions. Tanks (76 x 33 x 33 cm) filled with 75 l of filtered seawater at 29 salinity served as experimental units. Heaters regulated the water temperature at 25˚C and overhead LED lights were kept on a 12-h light cycle. To isolate TMII effects, toadfish or stone crabs were restricted within cylindrical (18 cm diameter x 30.5 cm tall) 2.6 cm mesh wire cages. Ten loose 15 mm SH oysters were distributed within each unit before the introduction of individual mud crabs belonging to one of three size classes: small (13–15 mm CW), medium (21–23 mm CW), or large (28–31 mm CW). The use of individual mud crabs precluded intraspecific interference and cues. Empty cage treatments controlled for cage effects. The substrate was withheld to limit responses of crabs solely to the presence of caged predators. Each of four randomized blocks consisted of three experimental units containing individual mud crabs of each size class in the presence of a caged predator (i.e., stone crab ≈ 65 mm carapace width; toadfish ≈ 120 mm TL), in addition to three control units containing individual mud crabs of each size class and empty cages. The duration of each run was 24 h, after which surviving oysters were enumerated. All runs for each type of top predator ensued over consecutive 4-d periods.

A second TMII experiment focused on interactions involving one large (28–30 mm CW) and three small (≈ 14–15 mm CW) mud crab predators, together with either caged toadfish (120 mm TL) or empty control cages. Experimental conditions were identical to those for individual mud crab TMII experiments. Runs ensued for 24 h, with the replacement of consumed oysters after 12 h to adjust for high feeding rates. Surviving oysters were enumerated at 12 and 24 h.

## Ethics statement

This study was carried out in strict accordance with the recommendations in the Guide for the Care and Use of Laboratory Animals of the National Institutes of Health. The protocol for this study was approved by the Institutional Animal Care and Use Committee (IUCUC) of the University of Southern Mississippi (Protocol Number: 16052606), based on an Animal Subjects Research Application submitted to the USM Office of Research Integrity. Also, VRS completed the Collaborative Institutional Training Initiative Program (CITI Program). Certificates

were obtained for both the animal subjects course and the common course covering various considerations related to ethical conduct in all professional and research activities. All efforts to obtain, handle, and house fish and invertebrate subjects were made humanely. Following the completion of experiments, fish subjects were released back into the wild. Field work was not conducted in protected or privately-owned areas, and no protected species were sampled. Collections of oyster reef animals were made under the auspices of a Scientific Research Permit issued to the USM Gulf Coast Research Laboratory by the Mississippi Department of Marine Resources (MS DMR).

## Statistical analysis

For MPE experiments, the mean numbers of oysters and mud crabs consumed were dependent variables within separate two-way ANOVAs with Treatment and Substrate as fixed factors using the General Linear Models procedure in SPSS (version 25). Tests of the blocking effect based on separate ANOVAs for each of the two substrate types in which Treatment was a fixed factor and Block was a random factor showed the blocking factor to be unimportant ($F = 2.116$, $P = 0.116$ oyster shell; $F = 1.701$, $P = 0.189$ limestone gravel). Numbers of both small and large oysters consumed were pooled as a single response, as the survival of both size classes was strongly correlated ($r^2 = 0.68$; $P_{2t} < 0.001$; $n = 60$) and significant differences in numbers of large vs. small oysters consumed were lacking across the MPE treatments (paired t-tests; Holm-Bonferroni sequential corrected rejection at $\alpha < 0.05$ [45]) (S1 Table). Within treatment differences in the mean numbers of small vs. large oysters consumed ranged from 0.4 to 1.9 (i.e., of the maximum possible difference of 15) across all treatments. The MPE control treatment (i.e., oysters only) was excluded from the GLM ANOVA to avoid an omnibus test that was overly influenced by the lack of oyster mortality for that treatment. Levene's and F-tests for unequal error variances and heteroscedasticity were not significant for oyster mortality within the MPE ANOVA. Follow-up LSD tests among treatments were interpreted based on Holm-Bonferroni sequential adjusted rejection levels [45]. The ANOVA for the mean number of mud crabs consumed consisted of the four MPE treatments involving mud crabs. Because error variance in the response was heterogeneous (Levene Statistic = 6.465; $P < 0.001$), mud crab mortality was tested relative to Treatment and Substrate within a two-way PERMANOVA using Manly's approach—Unrestricted Permutation of Observations in R.

For the first TMII experiment, the mean number of oysters consumed was examined as a response within a two-way ANOVA of Mud Crab Size (i.e., small, medium, large) crossed with Predator (i.e., stone crab (n = 12), toadfish (n = 12), no predator (n = 24)). Because the error variance in the response was heterogeneous (Levene Statistic = 4.925; $P < 0.001$), the ANOVA was conducted as a two-way PERMANOVA using Manly's approach. For the second TMII experiment involving the mixed-size mud crab treatment, an independent two-sample t-test compared the numbers of oysters consumed between the two groups (toadfish present vs. toadfish absent). Variance in the response was homogeneous between the two groups ($F = 0.866$; $P = 0.388$).

Reduction or enhancement of survival risk to oysters and mud crabs was examined for treatments from the MPE experiment using one-sample t-tests and a multiplicative model approach defined by Soluk and Collins [15] and Sih et al. [18]. Expected oyster mortality values were adjusted to account for oysters that would have been consumed by predated mud crabs (i.e., 1.464 oysters per mud crab based on ½ the number of mud crabs lost) [14]. No such adjustment was needed for mud crabs as predators, as mud crab cannibalism did not occur. Toadfish predation on oysters was assumed to be zero for calculations of expected numbers of oysters consumed. One-tailed t-tests reflected directionality in the expected outcomes, as

revealed by the relative difference between expected and observed values. For the MPE treatment involving all three predators (i.e., TF-SC-MC) for which oysters were prey, two different one-sample t-tests were conducted using expected values generated either from the three single predator outcomes or from the two top predators together and the mud crab only outcome. Also, for the MPE treatment involving all three predators (i.e., TF-SC-MC) for which mud crabs were prey, two different one-sample t-tests were conducted using stone crabs and toad-fish single predator outcomes either from both substrates pooled or from only limestone gravel substrate, which exhibited higher mud crab mortality. Where risk reduction was inferred (i.e., observed value less than expected value), consumptive effect (CE), and non-consumptive effect (NCE) proportions were calculated [14].

For the TMII experiments, non-consumptive effect (NCE) and consumptive effect (CE) proportions were obtained directly from the difference in oyster mortality between caged predator and empty cage treatments. The sparse design of TMII experiments ensured that observed NCEs were attributable exclusively to the presence of top predators. However, the MPE and TMII experiments were not directly comparable due to their incommensurate designs.

## Results

### MPE experiment

The mean number of oysters consumed differed for the MPE Treatment and Substrate factors, but not for their interaction (Table 1). Overall, more oysters were consumed in limestone gravel (15.6 ± 1.93 se) than in oyster shell (11.3 ± 1.27 se) across MPE treatments, except for the toadfish/mud crab (TF/MC) treatment for which five-fold more oysters were consumed in the oyster shell than in limestone gravel (t = 3.14; $P_{1-t}$ = 0.007) (Table 2; Fig 1). The largest between-treatment difference in the mean number of oysters consumed (i.e., 16.8) pooled across both substrates exemplified the strong inhibiting effect of toadfish on the consumption of oysters by mud crabs (Table 2). The observed number of oysters consumed for the TF/MC treatment (i.e., mean ± se = 3.7 ± 1.32) was significantly lower (i.e., Holm-Bonferroni sequential adjusted LSD tests) than that for the MC (P < 0.001), TF/SC (P = 0.003), SC/MC (P = 0.003) and TF/SC/MC (P = 0.001) treatments (Table 2). Comparable numbers of oysters were consumed across all treatments which involved stone crabs as one of the predators (mean ± se = 12.2 ± 2.89 SC; 14.1 ± 2.56 MC/SC; 14.1 ± 3.09 SC/TF). Thus, stone crab predation on oysters was not strongly inhibited by the presence of toadfish. Moreover, the combined effect of mud crabs and stone crabs on oyster mortality was not additive (mean ± se = 20.5 ± 1.65 MC; 14.1 ± 2.56 MC/SC), likely pointing to effects of inhibition of and predation on mud crabs by stone crabs. More oysters were consumed in the mud crab only treatment than in the stone crab only treatment (mean ± se = 20.5 ± 1.65 for 7 MC; 12.2 ± 2.89 for 1 SC).

The mean number of mud crabs consumed within the MPE experiment strongly differed for both main factors as well as for the interaction between MPE Treatment (i.e., the four MPE treatments with mud crabs) and Substrate (two-way PERMANOVA, MPE Treatment P = 0.0026; Substrate P < 0.001; MPE Treatment × Substrate P = 0.001). Except for the TF/MC treatment, mud crab loss did not occur within oyster shell substrate, nor for the MC treatment within limestone gravel substrate. Thus, mud crab cannibalism did not occur during the 24 h experimental period. Mud crab loss did occur for almost every treatment combination involving toadfish, except for the treatment comprising both top predators in oyster shell substrate.

The interaction effect within the two-way PERMANOVA reflected an uneven and higher loss of mud crabs in limestone gravel than oyster shell for the MPE treatments involving stone

**Table 1. Two-way ANOVA of the mean number of oysters consumed within the MPE experiment (bold P ≤ 0.05).** The corrected model term excludes variability explained by the intercept, thereby constituting an overall test of the dependent variables.

| Source | Type III Sum of Squares | df | Mean Square | F | P | Partial Eta Squared |
|---|---|---|---|---|---|---|
| Corrected Model | 2219.73 | 11 | 201.79 | 3.56 | **0.001** | 0.45 |
| Intercept | 10881.07 | 1 | 10881.07 | 191.79 | **< 0.001** | 0.80 |
| Substrate | 281.67 | 1 | 281.67 | 4.97 | **0.031** | 0.09 |
| MPE Treatment | 1547.33 | 5 | 309.47 | 5.46 | **< 0.001** | 0.36 |
| Substrate × Treatment | 390.73 | 5 | 78.15 | 1.38 | 0.249 | 0.13 |
| Error | 2723.20 | 48 | 56.73 | | | |
| Total | 15824.00 | 60 | | | | |
| Corrected Total | 4942.93 | 59 | | | | |

crabs (i.e., SC/MC and TF/SC/MC). Mud crab loss in limestone gravel averaged more than two-fold higher when the stone crab was the only top predator than when toadfish was the only top predator (mean ± se = 2.6 ± 1.03 vs. 1.0 ± 0.45). And mud crab loss was identical across both substrate types when toadfish was the only top predator (mean ± se = 1.0 ± 0.45

**Table 2. Numbers of oysters consumed for MPE experiment treatments for oyster shell and limestone gravel, as well as totals across treatments and substrates (maximum of 30 possible).**

| Substrate | MPE Treat | Mean | Std. Deviation | Std. Error | N |
|---|---|---|---|---|---|
| Oyster Shell | SC | 8.20 | 7.46 | 3.34 | 5 |
| | TF/SC | 8.40 | 3.05 | 1.36 | 5 |
| | MC | 17.80 | 5.54 | 2.48 | 5 |
| | SC/MC | 13.00 | 5.34 | 2.39 | 5 |
| | TF/MC | 6.20 | 4.32 | 1.93 | 5 |
| | TF/SC/MC | 14.20 | 9.31 | 4.16 | 5 |
| | Total | 11.30 | 6.96 | 1.27 | 30 |
| Limestone Gravel | SC | 16.20 | 9.60 | 4.29 | 5 |
| | TF/SC | 19.80 | 11.17 | 4.99 | 5 |
| | MC | 23.20 | 3.56 | 1.59 | 5 |
| | SC/MC | 15.20 | 10.76 | 4.81 | 5 |
| | TF/MC | 1.20 | 2.17 | 0.97 | 5 |
| | TF/SC/MC | 18.20 | 10.06 | 4.50 | 5 |
| | Total | 15.63 | 10.59 | 1.93 | 30 |
| Total | SC | 12.20 | 9.14 | 2.89 | 10 |
| | TF/SC | 14.10 | 9.78 | 3.09 | 10 |
| | MC | 20.50 | 5.23 | 1.65 | 10 |
| | SC/MC | 14.10 | 8.09 | 2.56 | 10 |
| | TF/MC | 3.70 | 4.17 | 1.32 | 10 |
| | TF/SC/MC | 16.20 | 9.38 | 2.97 | 10 |
| | Total | 13.47 | 9.15 | 1.18 | 60 |

MPE Treat = Multiple Predator Effect treatments.

SC = stone crab.

TF/SC = toadfish and stone crab.

MC = mud crab.

SC/MC = stone crab and mud crab.

TF/MC = toadfish and mud crab.

TF/SC/MC = toadfish, stone crab and mud crab.

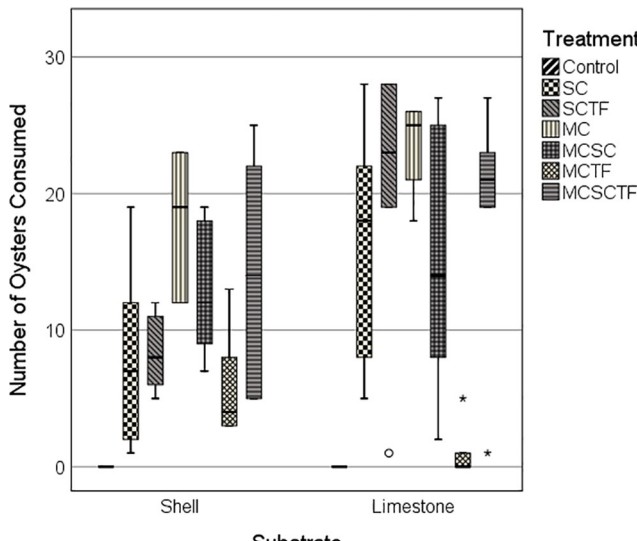

**Fig 1. Total number of juvenile oysters consumed across MPE treatments for limestone gravel and oyster shell substrates.** MPE treatments: Control = predators not present; SC = stone crab; TF/SC = toadfish and stone crab; MC = mud crab; SC/MC = stone crab and mud crab; TF/MC = toadfish and mud crab; TF/SC/MC = toadfish, stone crab and mud crab.

for both substrates). Accordingly, mud crab feeding was more inhibited by toadfish when it was the only top predator (oyster NCE = 0.81) than when the stone crab was the only top predator (oyster NCE = 0.37). And mud crab feeding was least inhibited in the presence of both top predators (oyster NCE = 0.25) (Table 3). As such, the greatest loss of mud crabs

**Table 3. Risk-model t-tests for MPE experiment treatments.**

| MPE Treat | Prey type | Exp N | MC N | Obsv N | SE Obs | t | $P_{1\text{-}t}$ | CE | NCE |
|---|---|---|---|---|---|---|---|---|---|
| TF+MC | Oyster | 19.04 | 20.50 | 3.70 | 1.32 | -11.645 | **<0.001** | 0.19 | 0.81 |
| SC+MC | Oyster | 24.36 | 22.46 | 14.10 | 2.56 | -3.268 | **0.005** | 0.63 | 0.37 |
| TF+SC | Oyster | 12.20 | NA | 14.10 | 3.10 | 0.614 | 0.277 | NA | NA |
| TF+SC+MC$_a$ | Oyster | 24.36 | 21.58 | 16.20 | 2.97 | -1.814 | **0.051** | 0.75 | 0.25 |
| TF+SC+MC$_b$ | Oyster | 16.06 | 13.28 | 16.20 | 2.97 | 0.985 | 0.175 | NA | NA |
| TF+SC+MC$_c$ | Mud crab | 3.23 | NA | 3.80 | 0.58 | 0.980 | 0.191 | NA | NA |
| TF+SC+MC$_d$ | Mud crab | 2.11 | NA | 1.90 | 0.69 | -0.310 | 0.381 | 0.90 | 0.10 |

MPE Treat = Multiple Predator Effect treatment; Exp N = expected number of prey; MC N = expected number of prey, corrected for lost mud crabs; Obsv N = observed number of prey; SE Obs = standard error of observed number prey; CE = consumptive effect; NCE = non-consumptive effect.

Negative t-values connote MPE risk reduction for basal prey; Positive t-values imply MPE risk enhancement for basal prey; 1-tailed t-tests imply expected directionality in the outcomes; bold P ≤ 0.05.

Correction for lost Mud Crabs (MC N) = 1.464 oysters per individual mud crab.

NCE not applicable for mud crabs as prey because mud crab cannibalism was not observed.

Toadfish predation on oysters assumed zero.

All tests for both substrate types combined, except for TF+SC+MCc = limestone only.

TF+SC+MC$_a$–toadfish, stone crab, and mud crab MPE relative to three single predator treatments.

TF+SC+MCb–toadfish, stone crab, and mud crab MPE treatment relative to the two top predators together treatment and the mud crab alone treatment.

TF+SC+MC$_c$–toadfish, stone crab, and mud crab MPE treatment for limestone gravel treatments.

TF+SC+MC$_d$–toadfish, stone crab, and mud crab MPE treatment for both substrates pooled.

CE and NCE are not applicable (NA) when Obs N > Exp N (i.e., positive t value).

(mean ± se = 3.80 ± 0.58) occurred in the presence of both top predators in limestone gravel. Moreover, mud crab loss was essentially additive relative to the two top predators, whether considering just limestone gravel (expected N = 3.23 vs. observed N = 3.80) or both substrates pooled (expected N = 2.11 vs. observed N = 1.90) (Table 3). The lack of much difference between the observed and expected loss of mud crabs in the presence of both top predators coincided with nonexistent or weak NCEs for mud crabs in limestone gravel or both substrates pooled, respectively (Table 3).

### TMII experiments

For the TMII experiment on mud crab size, the mean number of oysters consumed strongly differed for both main factors as well as for the interaction between Predator and Mud Crab Size (two-way PERMANOVA, Predator P < 0.001; Mud Crab Size P < 0.001; Predator × Mud Crab Size = 0.003). The interaction effect reflected non-parallel increases in numbers of oysters consumed with crab size across the three predator treatments (Fig 2). Markedly more oysters were consumed with increasing crab size within the empty cage treatment, whereas oyster consumption was greatly inhibited across the three crab sizes within the caged stone crab and toadfish treatments. Oyster consumption was roughly an order of magnitude higher in the absence of top predators: the mean number of oysters consumed across all size classes of mud crabs was 4.87 ± 0.31 se in empty cages vs. 0.58 ± 0.44 se and 0.17 ± 0.44 se in the presence of stone crab and toadfish, respectively. Oyster consumption by mud crabs appeared somewhat more inhibited by toadfish than by stone crabs (Fig 2). For the TMII experiment using mixed sizes of mud crabs, almost four-fold fewer oysters were consumed in the presence of caged toadfish (4.5 ± 2.1 se) than when cages were empty (16.8 ± 3.3 se) (t = 3.165; $P_{1t}$ = 0.009).

Non-consumptive effects (NCE) due to inhibition of oyster consumption varied with mud crab size and were less pronounced for stone crabs than for toadfish (Table 4). A noticeable trend in NCE strength with mud crab size was evident when data were pooled for both top predators (Table 4). Large mud crabs showed the lowest NCE values, although NCEs were relatively high for all three size classes (i.e., NCEs > 0.79). The NCE value for the mixed-size TMII experiment was lower than that for the large mud crab treatment from the TMII experiment on mud crab size in the presence of toadfish (NCE = 0.73 vs. 0.93), implying interference among mud crabs did not appreciably inhibit the consumption of oysters, even in the absence of substrate.

## Discussion

Studies of MPEs within oyster reef assemblages usually focus on how interspecific interactions affect the stability of the oyster reef tri-trophic cascade through non-additive predator effects [15]. In addition to predator identity [9], other ecological factors that mediate the stability of the oyster-reef trophic cascade include body size [23], predator density [46], and habitat complexity or patchiness [14, 47]. As such, the stability of a given oyster-reef configuration is evaluated within the context of limited combinations of ecological factors [9, 36]. Even given its uncomplicated structure, studying the multitude of trophic interactions within the entire oyster-reef food web together would be unwieldy. So, experimental factors are selected judiciously to obtain new insights into oyster-reef trophic stability. Here, we chose to examine the tri-trophic toadfish-mud crab-oyster cascade in connection with the interactive role played by the stone crab, an intraguild predator, and common resident of Gulf of Mexico (GoM) oyster reefs.

As often shown in other studies, the toadfish was a primary stabilizing influence on the tri-trophic cascade in our study. However, unlike Grabowski et al. [14], where the tri-trophic

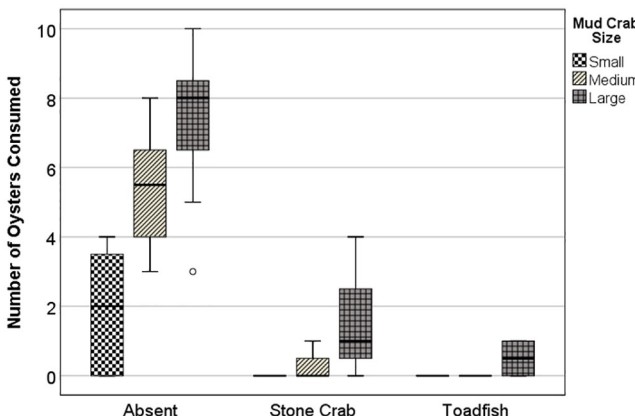

**Fig 2. TMII experiment.** Consumption of juvenile oysters by three size classes of mud crabs across three caged predator treatments, absent, stone crab, and toadfish.

cascade was reinforced by interactions between toadfish and the generalist blue crab capable of consuming both intermediate predators and basal oyster prey, the addition of the intraguild stone crab did not reinforce the tri-trophic cascade in our study. Toadfish did not inhibit stone crab feeding. Similarly, generalist predators have been shown to destabilize oyster reef trophic cascades [48]. In addition to suppressing feeding by mud crabs on oysters through both direct consumption of mud crabs and inhibition of mud crab feeding behavior, stone crabs directly consumed oysters as a shared prey. However, mud crab feeding behavior was not as strongly inhibited by stone crabs as by toadfish in the MPE experiment. Further weakening of the trophic cascade was indicated by less inhibition of mud crab feeding when all three predators were together (NCE = 0.25) than when toadfish (NCE = 0.81) or stone crabs (NCE = 0.37) occurred separately with mud crabs. Moreover, the trophic cascade was not stabilized by intraspecific interference and cannibalism among mud crabs, as in Geraldi [23]. Thus, the trophic balance was generally destabilized when all three predators were together, as compared to when toadfish and mud crabs were the only two predators.

**Table 4. Consumptive effects (CE) and non-consumptive effects (NCE = 1 −CE) for caged predator TMII experiments.** Predator present vs. predator absent values represent numbers of 15 mm oysters consumed during the experiments (out of 10 possible except for the mixed size treatment, which used 20 oysters).

| Treatment | Predator Present | Predator Absent | CE | NCE |
|---|---|---|---|---|
| SC + MC_L | 1.50 | 7.375 | 0.203 | 0.797 |
| SC + MC_M | 0.25 | 5.375 | 0.046 | 0.953 |
| SC + MC_S | 0.00 | 1.875 | 0.000 | 1.000 |
| TF + MC_L | 0.50 | 7.375 | 0.068 | 0.932 |
| TF + MC_M | 0.00 | 5.375 | 0.000 | 1.000 |
| TF + MC_S | 0.00 | 1.875 | 0.000 | 1.000 |
| BOTH+MC_L | 1.00 | 7.375 | 0.136 | 0.864 |
| BOTH+MC_M | 0.125 | 5.375 | 0.023 | 0.977 |
| BOTH+MC_S | 0.00 | 1.875 | 0.000 | 1.000 |
| TF + MC_{L/S} | 4.50 | 16.75 | 0.269 | 0.731 |

BOTH = data pooled for caged toadfish and stone crab.

L, M, and S = large, medium, and small mud crab.

L/S = mixed size mud crab TMII experiment.

Toadfish stabilized the trophic cascade mainly by inhibiting mud crab feeding on oysters rather than by directly consuming mud crabs. Indeed, direct predation on mud crabs was higher for stone crab than for toadfish in limestone gravel. But oyster consumption by mud crabs was not inhibited as much by stone crabs as by toadfish, as shown by a more than two-fold difference in NCEs within the MPE experiment. Nevertheless, the combined effect of predation on and inhibition of mud crabs by stone crabs carried over to a nonadditive effect of fewer oysters consumed than expected when both intraguild crabs were together. In contrast, consumptive effects of toadfish and stone crabs on mud crabs as prey were additive when all three predators were together. Previous studies provide some insights. Toadfish can induce mud crabs to seek refuge and limit their movements [14]. Stone crabs are more adept at flushing mud crabs from refuges, whereas toadfish are ambush predators [47]. Consequently, predation on mud crabs was highest in the presence of both top predators. The disposition of stone crabs as a stabilizing influence on the trophic balance may hinge on the profitability of oysters and mud crabs as prey. Both mud crabs and oysters were consumed by stone crabs in our study. Depending on the costs and benefits as a function of predator-prey size vs. handling time, stone crabs might prefer mud crabs or oysters as more profitable prey types, but this needs to be tested.

It was expected that toadfish might inhibit stone crab feeding behavior because as common residents of GoM oyster reefs juvenile stone crabs regularly occur and can even surpass mud crabs in toadfish diets [41]. However, toadfish did not inhibit the consumption of mud crabs by stone crabs in our MPE experiment. Nor did toadfish inhibit the consumption of oysters by stone crabs in the absence of mud crabs. The sizes of stone crabs used in our MPE experiment matched the most common size class that can consume adult mud crabs. Even though the total lengths of toadfish (i.e., 120 mm TL) were about two-fold greater than the carapace widths of stone crabs, stone crabs may have been too large to perceive much risk of toadfish predation within the MPE experiment. However, provided toadfish can grow large enough (i.e., > 300 mm TL) [23, 49] to pose a predatory risk to adult stone crabs in the field, crabs of the size used in our experiment may not have entirely outgrown the risk of predation by toadfish. Indeed, the large mouth width of the oyster toadfish equals 20% of its TL [49].

Body size is an important ecological factor mediating the stability of food webs through predator-prey interactions between the top and intermediate predators [50] and between intermediate or intraguild predators and basal prey [20, 43, 48]. Body size also mediates interference or predation among intraguild predators [22, 23]. For example, the size-selection of hard clams (*Mercenaria*) proved to be important for understanding the preference of small clams by blue crabs but size-selection was not important for stone crabs with strong crushing ability. Notwithstanding the disparate body sizes of intermediate mud crabs and intraguild stone crabs, the size selection of oysters was not very strong for either crab species in our MPE experiment. Indeed, the ratio of small (i.e., 15 mm spat) to large (i.e., 30 mm seed) oysters consumed by stone crabs vs. mud crabs when alone was 45%: 55% and 55%: 45%, respectively. Additionally, the relative numbers of small vs. large oysters consumed were not significantly different across all of the MPE treatments. However, the use of loose oysters rather than oysters attached within a reef matrix as well as the narrow size range of sizes of oysters limits the applicability of our results. When considered over a wider range of sizes (i.e., up to 70 mm valve length (VLV)), costs associated with handling time or crushing success of hard shell molluscan prey increases with the predator-prey size ratio, and as such can affect the feeding preferences of stone crabs, blue crabs, and mud crabs [43]. A size threshold below 25 mm VLV served as a refuge from stone crab predation [43]. Conversely, small oysters below 25 mm VLV were vulnerable to predation by mud crabs. Stone crabs used in that study were larger (80–130 mm CW) than those used in our MPE experiment (60 mm CW). And mud crabs (23 mm CW)

readily consumed oysters in both size classes in our experiment. Indeed, both stone crabs and mud crabs possess strong crushing chelae.

Modest size-related NCE release was detected for large mud crabs in the TMII experiment, as shown by higher oyster consumption by large mud crabs compared to smaller mud crabs in the presence of top predators. Plasticity in the expression of TMIIs imposes an important influence on trophic cascades [13]. Different sizes of intermediate predators should respond commensurately with the perceived threat of predation, as a balance between risk and energetic trade-offs [6]. Thus, TMII strength can vary with the intermediate predator size [26], and the size structure of the intermediate predator population mediates predation pressure on prey populations. Risk perception by intermediate predators may be influenced by top predator body size [22] and top predator biomass where aqueous chemical cues are involved [55]. However, Geraldi [23] noted the lack of a definite predator-prey size relationship between mud toadfish and crabs, which he attributed to the large mouth and ambush feeding strategy of toadfish. Thus, the observed NCE release for large mud crabs in our study is likely a general response to predation risk.

The observed size-related NCE release for mud crabs in our study complements the finding by Geraldi [23] that the oyster-reef trophic cascade was mediated by the size composition and density of intermediate mud crabs. But in contrast to our study, the trophic cascade was stabilized by interference and cannibalism between mud crabs in the Geraldi study. Surprisingly, mud crab interactions exerted as much influence on trophic stability as did the inhibition of mud crab feeding by the presence of a top predator [23]. Neither interference nor cannibalism among mud crabs played an important role in our study. However, a much wider size range of mud crabs was exposed over a longer experimental period in the Geraldi study. Consequently, cannibalism was focused on small mud crabs, owing to the inverse relationship between molting frequency (i.e., vulnerability) and crab size. The 24-h period for our MPE experiment precluded molting frequency as a factor. Furthermore, density mediated interference between mud crabs was likely fostered by using biomass equivalents to represent small-bodied mud crabs in the Geraldi study [23]. Representing oyster reef predators by their biomass equivalents often reveals density mediated predation on oysters [14, 23]. As such, the exclusion of abundant mud crabs promotes oyster survival in the field [40, 48]. Thus, interference and cannibalism among mud crabs will stabilize the trophic cascade, whereas size-related NCE release for mud crabs could weaken the trophic cascade. Additionally, oyster consumption was not inhibited by interference among mud crabs in our mixed size TMII experiment.

Through the provision of spatial refuge, habitat complexity can affect the stability of oyster-reef trophic cascades by buffering predation on basal prey [14] and intermediate predators [22], or by mitigating interference [29, 36]. Due to its three-dimensional structure containing many crevices, oyster shell offers greater habitat complexity than limestone gravel. Surface area to volume ratios of exposed oyster shell range 3- to 9-fold higher than exposed limestone gravel under field conditions [51]. As a result, the consumption of juvenile oysters averaged 1.4-fold higher in limestone gravel than in oyster shell in our study, demonstrating the general stabilizing effect of complex habitat [14]. And equal numbers of oysters in both size classes were consumed across both types of substrate (51% small vs. 49% large in oyster shell; 50% small vs. 50% large in limestone gravel). Presumably, oysters were easier to find in limestone gravel than oyster shell. In contrast, the stabilizing effect of toadfish within the trophic cascade was weakened somewhat by habitat complexity. In the presence of toadfish, the consumption of oysters by mud crabs was 5-fold higher in the oyster shell than in limestone gravel, where mud crabs would have been more vulnerable to predation. Conceivably, the availability of more refuge space for mud crabs facilitated their ability to actively feed within the complex oyster shell substrate. In contrast, Grabowski et al. [14] found that the trophic cascade was

stabilized by the availability of refuge space for mud crabs which were less active in the presence of both toadfish and blue crabs together. Likewise, oyster consumption was lower in oyster shell than in limestone gravel when mud crabs were present with both top predators in our study. Other studies have shown that habitat complexity can destabilize the tri-trophic cascade by reducing interference involving intermediate and top predators [29, 36]. Thus, the stabilizing influence of habitat complexity on the oyster trophic cascade is context-dependent. Effects of habitat complexity are particularly important considering oyster-reef restoration practices, which often entail the use of limestone gravel as substrate [51–54].

While our MPE experiment roughly simulated oyster reef habitat, it was less realistic than natural conditions in several respects. Selected taxa represented a small subset of the oyster reef community. Consequently, predators were precluded from having mixed diet choices [14]. Predators were restricted to narrow size ranges. Although densities of predators were analogous to the field, they did not vary across treatments. Temperature and salinity were maintained at constant levels and did not fluctuate. Also, the lack of turbid conditions potentially altered perceptual modalities used by predators and prey [55]. The presentation of oysters as prey was also unnatural. Predation efficiency likely differs between loose oysters and settled attached oysters due to prey handling differences. Survival is lower for cultchless (i.e., lacking settlement substrate) oysters than for attached oysters in the field [40]. Although two size classes of oyster prey were used, the juvenile oyster size distribution was not fully represented. Our experiment also incurred some prey depletion, although full depletion was precluded by calibration based on pilot runs. Biological processes acting over periods longer than the 24-h experimental period were also precluded, as noted for molting above. Survival of oysters and mud crabs would likely be different in oyster shell substrate consisting of loose whole dead oyster shells than within the natural reef matrix. Additionally, restricted movements of experimental animals possibly altered predation efficiency and increased chances of interference within experimental units. Finally, the isolation of experimental units likely enhanced NCEs relative to open field conditions involving currents by concentrating chemical cues [55]. However unrealistic, our experiments still provided useful insights into processes potentially operating in the field [55].

The MPE and TMII experiments were complimentary in ways that provided insights into the expression of NCEs. The magnitude of NCEs varies with the perception of risk as a function of many mediating factors [55], including some that differed between the MPE and TMII experiments. The TMII experiment using individual mud crabs was devised to exclude all mediating factors other than cues emanating from top predators. Compared to the MPE experiment, NCEs within the TMII experiment were differentially influenced by higher concentrations of chemical cues from top predators due to caging stress and smaller water volumes, stronger visual and auditory cues presented by top predators, the lack of chemical cues from other injured or living mud crabs, the absence of interference from conspecifics or top predators, the lack of cues produced by other animals, and greater sensitivity to perceived risk in the absence of substrate. Greater risk sensitivity in the TMII experiment seems likely, as mud crabs were less risk-averse in oyster shell than in limestone gravel in the MPE experiment. Also, mud crab feeding was greatly inhibited in the presence of top predators in the TMII experiment, despite oysters being more readily available than in the MPE experiment. Indeed, the huge disparity between the high consumption of oysters by mud crabs in the absence of both predators and substrate and the extremely low consumption in the presence of predators notwithstanding the lack of substrate in the TMII experiment may largely explain the difference in NCEs between the MPE and TMII experiments. Feeding by intermediate predators was inhibited in the presence of top predators in both the MPE and TMII experiments, but feeding was even more inhibited, and the relative difference in NCEs between top predators

was less in the TMII setting. Still, the TMII experiments confirmed strong NCEs attributable strictly to the presence of the top predators in the absence of other mediating factors.

## Conclusions

Laboratory experiments revealed several potentially destabilizing effects on the tri-trophic oyster reef cascade that complement previous studies and warrant further testing. First, the intra-guild stone crab consumed oysters and mud crabs undeterred by the presence of toadfish. Feeding on oysters by mud crabs was inhibited by the presence of both toadfish and stone crabs, but considerably less so by the latter in the MPE experiment. Consequently, the total effect of all three predator species together weakened the tri-trophic cascade; substantially fewer oysters were consumed when toadfish and mud crabs were the only two predators. While feeding by mud crabs was inhibited in the presence of both top predators together, inferred NCEs in terms of oyster loss were stronger for each top predator individually. Weaker inhibition of mud crab feeding was inversely related to higher direct predation on mud crabs by stone crabs compared to toadfish. The TMII experiments showed that strong NCEs were attributable strictly to the inhibition of mud crab feeding by top predators in the absence of other mediating factors, and inferred that the inhibition of mud crab feeding could largely account for NCEs within the MPE experiment, rather than interference involving other mud crabs or top predators. An inverse relationship between mud crab body size and NCE strength observed in the TMII experiment presents an additional potentially destabilizing influence on the tri-trophic-cascade. Finally, although habitat complexity generally stabilized the trophic cascade across MPE treatments, greater habitat complexity mitigated the inhibition of mud crab feeding in the presence of toadfish. Still, the toadfish-mud crab-oyster configuration exhibited the lowest oyster consumption, regardless of the substrate. Although multiple studies confirm the efficacy of the tri-trophic cascade, some have questioned its relevance under field conditions [40] or have shown that some ecological interactions can weaken it [19, 22, 36, 48]. This study underscores how certain ecological interactions can mediate trophic cascades within variable environments. Our findings must be considered tentative as they emanate from experiments conducted under highly controlled laboratory conditions within a simulated habitat. However, this study provides additional insights into the trophic dynamics of oyster reefs for further testing under natural conditions.

## Supporting information

**S1 Table. Small vs. large oysters consumed across MPE treatments.** Paired t-tests between numbers of small vs. large oysters consumed across treatments for the MPE experiment. N = 10 for every treatment; N = 5 each representing each substrate type within each treatment. Rejection P = the Holm-Bonferroni sequential adjusted value for that comparison. No test is significant at familywise 0.05 level based on adjusted P. Mean differences (i.e., absolute value) range from 0.4 to 1.9 out of a maximum of 15 possible (i.e., 15 small and 15 large) across treatments.
(DOCX)

## Acknowledgments

This paper is based on research conducted by VRS in fulfillment of the Ph.D. degree in Coastal Sciences from the University of Southern Mississippi, U.S.A. We express our sincere gratitude to M. LaPeyre, R. Leaf, R. Hendon, and K. Dillon for their valuable advice and assistance as Ph. D. committee members. Many thanks are due to S. Rikard and the Auburn Shellfish

Laboratory at Dauphin Island for providing oyster spat for this study, and to C. Lapniewski and S. McIntosh, for their help with fieldwork. Finally, we thank two anonymous reviewers and the our academic editor for their valuable insights.

## Author Contributions

**Conceptualization:** Virginia R. Schweiss, Chet F. Rakocinski.

**Data curation:** Virginia R. Schweiss.

**Formal analysis:** Chet F. Rakocinski.

**Funding acquisition:** Virginia R. Schweiss, Chet F. Rakocinski.

**Investigation:** Virginia R. Schweiss.

**Methodology:** Virginia R. Schweiss, Chet F. Rakocinski.

**Project administration:** Virginia R. Schweiss, Chet F. Rakocinski.

**Resources:** Virginia R. Schweiss, Chet F. Rakocinski.

**Supervision:** Virginia R. Schweiss, Chet F. Rakocinski.

**Validation:** Virginia R. Schweiss.

**Visualization:** Chet F. Rakocinski.

**Writing – original draft:** Virginia R. Schweiss.

**Writing – review & editing:** Virginia R. Schweiss, Chet F. Rakocinski.

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
