## [Decision Letter · Decision Letter 0]

17 Jun 2020

PONE-D-20-05873

Destabilizing effects on a classic tri-trophic oyster-reef cascade

PLOS ONE

Dear Dr. Rakocinski,

Thank you for submitting your manuscript to PLOS ONE. After careful consideration, we feel that it has merit but does not fully meet PLOS ONE’s publication criteria as it currently stands. Therefore, we invite you to submit a revised version of the manuscript that addresses the points raised during the review process.

We look forward to receiving your revised manuscript.

Kind regards,

Romuald N. Lipcius, Ph.D.

Academic Editor

PLOS ONE

Additional Editor Comments:

Dear Dr. Rakocinski,

We have received two reviews of your manuscript (ms). As you will see from the comments made by the reviewers, both felt that the ms may be suitable for PLOS ONE after revision. I agree and ask that you submit a revised version of the ms.

Please carefully consider the comments of the reviewers and provide a point-by-point response which clearly and explicitly defines the changes made to satisfy each comment/criticism or argues your point, thoroughly and convincingly, in response to any comment with which you disagree. Once you submit a revision, I will re-evaluate it and your responses to the reviewers. The revision must deal with the specific comments of the reviewers. Please refer to line numbers of the revised manuscript whenever possible in your response letter, so that your changes can be easily assessed. When applicable, make sure to address reviewers' questions and comments not only in your response letter but also in the manuscript, since other readers will likely have the same questions.

Sincerely,

Romuald N. Lipcius

Journal Requirements:

2. In your Methods section, please provide additional location information of the collection sites, including geographic coordinates for the data set if available.

4. Your ethics statement must appear in the Methods section of your manuscript. If your ethics statement is written in any section besides the Methods, please move it to the Methods section and delete it from any other section. Please also ensure that your ethics statement is included in your manuscript, as the ethics section of your online submission will not be published alongside your manuscript.

Reviewers' comments:

Reviewer's Responses to Questions

**Comments to the Author**

1. Is the manuscript technically sound, and do the data support the conclusions?

Reviewer #1: No

Reviewer #2: Yes

2. Has the statistical analysis been performed appropriately and rigorously? 

Reviewer #1: Yes

Reviewer #2: I Don't Know

3. Have the authors made all data underlying the findings in their manuscript fully available?

Reviewer #1: Yes

Reviewer #2: Yes

4. Is the manuscript presented in an intelligible fashion and written in standard English?

Reviewer #1: Yes

Reviewer #2: Yes

5. Review Comments to the Author

Reviewer #1: This is an interesting and potentially valuable study that examines the stability of behavioral and density mediated interactions in an important and well-studied system. The study is adequately grounded in the context of current knowledge, with a series of carefully and properly analyzed studies that, together, attempt to examine a number of important variables known or suspected to influence the stability of trophic cascades. A particular strength of the approach is the use of separate experiments designed to elucidate whether multiple predator effects (MPE) are the result of direct interactions between predators or changed patterns of behaviorally mediated indirect interactions caused by changed responses of mud crabs to perceived risk. Unfortunately, this is also a significant source of problems since the non-parallel nature of the two experiments prevents the authors from accurate inferences about the role of behavior in producing the effects. This reviewer sees little way the two experiments can be directly related to one another. This undercuts the results discussed in lines 335-341 and Table 6, and the issues discussed on Lines 373-393 that contributes to this study’s novelty.

Prey responses to risk depend on the intensity, frequency and salience of the risk cue and the presence of potential refuges (among other variables). This is the result of decades of work and has been extensively reviewed both generally and specifically with respect to chemical signals; work by Dill, Lima, Weissburg, Katz, Chivers, G Brown, Ferrari, Relyea are good and recent examples. Thus, these variables must be at least coarsely controlled in order for the results from the two different experiments to be combined to evaluate the role of consumption and predator interactions vs. mud crab behavioral responses. The present study results in different experimental conditions affecting risk cues used by mud crabs and risk perception by mud crabs, which greatly complicates the interpretation of the results to the point that the conclusions are unreliable. It’s not even clear how to argue logically about the nature and direction of the various discordances.

The specific problems are:

a. The use of different water volumes that alter the concentration of chemical risk cues from predators (L136 vs 155).

b. The MPE experiments will result in the production of additional risk cues from injured conspecifics that are lacking in the experiments with caged predators (L157 ) incapable of consuming mud crabs (MC). Moreover, predators in the MPE experiments are consuming MCs and oysters that result in additional or different risk cues available to MCs in these experiments. Hill, Smee, Weissburg have shown that both predator diet and biomass consumed result in risk cues with different salience to MC and oysters.

c. The use of caged vs. free roaming predators creates different potential risk cues. Okuyama and Bolker advocate for non-lethal predators that can interact but not consume prey as the best way to separate consumptive vs. non-consumptive effects. The present analysis presented in Table 6 to calculate these effect sizes is not reliable. Hill has shown that mud crabs respond to chemical as well as other stimuli (visual or auditory) when evaluating predator risk.

d. The use of different substrates may alter MC tolerance to risk and therefore affect their “calculation” of perceived risk

Accurately accounting for the role of mud crab behavioral responses to risk cues would be an important and novel

contribution, but would require a different experimental design. The use of equal water volumes and substrates, and caged predators that have consumed mud crabs and oysters as in the MPE experiments would alleviate some of the concerns. Allowing the predators to roam free while restricting their ability to consume prey would be an even better design; banding the stone crab claws certainly is possible and sewing closed the mouth of the toad fish may be possible.

The present experiments using caged predators still provides insights into effects of intermediate predator size since these results are not as directly together (i.e. Table 6).

There are a number of other stylistic, wording, and presentation suggestions on the marked PDF

Reviewer #2: The authors selected an interesting system that is amenable to manipulative experimental approaches. I liked their approach of combining experiments on direct predation by mud crabs on oysters with experiments to examine indirect effects of top predators on mud crab predation. Unfortunately, the main message gets lost amid much ecological jargon and statistics-speak. I encourage the authors to strive for simple, clear language. In particular, the Abstract should be written to appeal to a non-specialist audience. It currently contains a lot of jargon and acronyms, some of which (e.g., MPE) are not well defined. The last sentence of the Abstract, which is identical to the last sentence of the Conclusion, does not leave the reader with a clear take-home message. On a more positive note, I found the Introduction section to be more readable.

Methods:

1) Could using loose oysters rather than clumps of oysters have resulted in higher rates of predation?

2) In the TMII experiments, toadfish 120 mm in TL were confined in cages of 180 mm diameter. Could this have caused the fish to become stressed and release chemicals that could have influenced crab behavior?

3) Also in the TMII experiments, no substrate was used, which is far from a realistic situation and could have affected mud crab behavior. The lack of a complex substrate combined with the smell of predators could have greatly reduced feeding by mud crabs.

4) The statistical analyses were complex and I do not feel qualified to evaluate them.

Results:

This section is difficult to read, possibly because the authors focused on the results of the statistical tests rather than the biological results. I would suggest focusing on the latter and using the former to bolster statements.

Table 3: One p value is 0.051 and should not be presented in bold font.

It might be useful to show the data on predation of mud crabs by stone crabs and toadfish in a figure.

Discussion:

Lines 369-70: The stone crabs used were relatively large compared to the toadfish, and that might be the reason that the fish didn’t affect stone crab feeding.

Lines 386-88: It doesn’t appear in Fig. 2 that toadfish had a much higher effect on mud crab behavior than stone crabs. Moreover, the lack of a substrate could have affected mud crab feeding behavior.

Last sentence (lines 488-90): I would argue that sacrificing realism does not facilitate understanding of natural mechanisms.

6. PLOS authors have the option to publish the peer review history of their article (what does this mean?). If published, this will include your full peer review and any attached files.

Reviewer #1: No

Reviewer #2: No

---

## [Author Response · Author response to Decision Letter 0]

31 Jul 2020

Response to Reviewers and Academic Editor – PONE-D-20-05873

We thank the anonymous reviewers and academic editor for their expert and constructive comments on manuscript PONE-D-20-05873. In response, we made every effort to address the concerns and implement the suggestions. We also revised the manuscript extensively in view of the criticisms and with the objective of reaching PLOS ONE publication standards. Hopefully, we have succeeded. The following details our responses to the concerns and suggestions. 

Comments from the editor and reviewers:

General Comments from the Academic Editor

1) Please carefully consider the comments of the reviewers and provide a point-by-point response which clearly and explicitly defines the changes made to satisfy each comment/criticism or argues your point, thoroughly and convincingly, in response to any comment with which you disagree.

See this document.

 Acknowledged.

b. In your Methods section, please provide additional location information of the collection sites, including geographic coordinates for the data set if available.

We now provide explicit location data for sources of experimental animals – including numbers and kinds of specific sites and a general geographic range of GPS coordinates (see Lines 137-141 in ‘Manuscript’). More detailed coordinates for the specific sites are available if required.

c. Should your manuscript be accepted for publication, we will hold it until you provide the relevant accession numbers or DOIs necessary to access your data. If you wish to make changes to your Data Availability statement, please describe these changes in your cover letter and we will update your Data Availability statement to reflect the information you provide.

After consulting the options PLOS ONE provides for data sharing, we chose to engage with the DRYAD digital repository for our data. And we have made note of that in the cover letter as advised. The data availability statement has also been updated accordingly within the revised manuscript (see Lines 767-773 in ‘Manuscript’).

d. Your ethics statement must appear in the Methods section of your manuscript. If your ethics statement is written in any section besides the Methods, please move it to the Methods section and delete it from any other section. Please also ensure that your ethics statement is included in your manuscript, as the ethics section of your online submission will not be published alongside your manuscript.

Upon this request we have moved our ethics statement to the Materials and Methods section under its own heading and following a description of the experimental methods (see Lines 197-211 in ‘Manuscript’).

e. While revising your submission, please upload your figure files to the Preflight Analysis and Conversion Engine (PACE) digital diagnostic tool. 

We have taken advantage of the useful PACE portal to create publication quality figures for the revision. The source figures were also modified as recommended by one reviewer.

Reviewer 1

Responses to General comments

1) the non-parallel nature of the two experiments prevents the authors from accurate inferences about the role of behavior in producing the effects. This reviewer sees little way the two experiments can be directly related to one another. 

It was not our intention to directly relate the MPE and TMII experiments without acknowledging differences in their designs. Nevertheless, reviewer 1 provided a good critique along with some additional interpretations and key references we incorporated into the revision. We now acknowledge the incommensurate nature of the experiments in several places with caveats within the revised manuscript and develop a new paragraph within the discussion that contrasts differences between the MPE and TMII experiments as a way to gain insights into potential factors mediating the expression of NCEs. (See Abstract line 38; Methods Line 172; Lines 264-265; Lines 534-555 in ‘Manuscript’).

2) different experimental conditions affecting risk cues used by mud crabs and risk perception by mud crabs, which greatly complicates the interpretation of the results to the point that the conclusions are unreliable

This comment also refers to interpretation of the TMII experiment vis a vis the MPE experiment. We acknowledge the incommensurate nature of the experiments in several places within the revised manuscript and develop a new paragraph within the discussion contrasting differences between the MPE and TMII experiments as a way to gain insights into potential factors mediating the expression of NCEs. (See Abstract line 38; Methods Line 172; Lines 264-265; Lines 534-555 in ‘Manuscript’). 

3) experiments using caged predators still provides insights into effects of intermediate predator size since these results are not as directly together

This comment by reviewer 1 appears to acknowledge the value of the size factor within the TMII experiment – which we appreciate.

4) edits on manuscript by reviewer 1 were followed and addressed:

Table 3: legend modified extensively for clarity as indicated by the reviewer and to aid interpretation. Legends of the other tables also edited to conform to PLOS ONE formatting.

2nd paragraph of the Discussion – statement rewritten to clarify the target of NCEs (e.g., Lines 404 -408 and elsewhere) 

3rd paragraph of Discussion – clarified NCE recipient. (Lines 413-415) 

8th paragraph of Discussion – clarified that experiments roughly simulated oyster habitats. (Lines 513-514)

8th paragraph of Discussion – acknowledged MC survival could be better under more natural refuge scenario (Lines 519-522)

9th paragraph of Discussion – new paragraph dedicated to contrasting the MPE and TMII considering incommensurate differences in their designs (Lines 534-555)

End of conclusions and abstract closing with nuanced statement re how the findings provide several insights worth testing under more natural conditions (Lines 46-48; 581-584)

Figures have been revised using combination shading/pattern scheme as suggested by reviewer 1

Reviewer 2

Responses to general comments

1) Unfortunately, the main message gets lost amid much ecological jargon and statistics-speak. I encourage the authors to strive for simple, clear language. In particular, the Abstract should be written to appeal to a non-specialist audience. It currently contains a lot of jargon and acronyms, some of which (e.g., MPE) are not well defined. The last sentence of the Abstract, which is identical to the last sentence of the Conclusion, does not leave the reader with a clear take-home message.

The abstract has been rewritten for clarity using less ecological jargon and a clear take-home message.

2) Could using loose oysters rather than clumps of oysters have resulted in higher rates of predation?

5th and 8th paragraphs of Discussion – artificial aspects of oyster presentation have been addressed more fully in the discussion (Lines 457-459; Lines 520-522)

3) In the TMII experiments, toadfish 120 mm in TL were confined in cages of 180 mm diameter. Could this have caused the fish to become stressed and release chemicals that could have influenced crab behavior?

This point is acknowledged as one of the ways the expression of NCEs may have been influenced differentially in the discussion (Lines 539-540)

4) Also in the TMII experiments, no substrate was used, which is far from a realistic situation and could have affected mud crab behavior. The lack of a complex substrate combined with the smell of predators could have greatly reduced feeding by mud crabs.

This point is also acknowledged as one of the ways the expression of NCEs may have been influenced differentially in the discussion (Lines 547-550 and elswhere)

5) Results section is difficult to read, possibly because the authors focused on the results of the statistical tests rather than the biological results. I would suggest focusing on the latter and using the former to bolster statements 

The Results section has been extensively rewritten to focus on biological results and place the statistical results in a more supportive context – in so doing, some redundant analyses were eliminated and as well as a couple tables. 

6) The stone crabs used were relatively large compared to the toadfish, and that might be the reason that the fish didn’t affect stone crab feeding.

This point is discussed more fully relative to the MPE experiment. We also now make the point that the size of stone crab had to be large enough to be able to consume mud crabs (Lines 427-437) 

7) It doesn’t appear in Fig. 2 that toadfish had a much higher effect on mud crab behavior than stone crabs. Moreover, the lack of a substrate could have affected mud crab feeding behavior.

These points are more fully recognized and interpreted accordingly within the revision (e.g., Lines 547-553)

8) I would argue that sacrificing realism does not facilitate understanding of natural mechanisms.

We agree and now close with more nuanced conclusions stating how the findings provide several insights worth testing under more natural conditions (Lines 46-48; 581-584)

---

## [Editor Report · Decision Letter 1]

12 Aug 2020

PONE-D-20-05873R1

Destabilizing effects on a classic tri-trophic oyster-reef cascade

PLOS ONE

Dear Dr. Rakocinski,

Thank you for submitting your manuscript to PLOS ONE. After careful consideration, we feel that it has merit but does not fully meet PLOS ONE’s publication criteria as it currently stands. Therefore, we invite you to submit a revised version of the manuscript that addresses the points raised during the review process.

In the attached edited version of the revised manuscript, I have added final comments and changes to be made.

We look forward to receiving your revised manuscript.

Kind regards,

Romuald N. Lipcius, Ph.D.

Academic Editor

PLOS ONE

Additional Editor Comments (if provided):

See attached edited manuscript.

---

## [Author Response · Author response to Decision Letter 1]

19 Sep 2020

2nd Revision Responses to Academic Editor – PONE-D-20-05873

We thank the academic editor for the expert and constructive comments on manuscript PONE-D-20-05873. In response, we made every effort to address the concerns and implement the suggestions. We also revised the manuscript with the objective of reaching PLOS ONE publication standards. Hopefully, we have succeeded. The following details our responses to the concerns and suggestions. 

Comments from the editor:

General Comments from the Academic Editor

Thank you for submitting your manuscript to PLOS ONE. After careful consideration, we feel that it has merit but does not fully meet PLOS ONE’s publication criteria as it currently stands. Therefore, we invite you to submit a revised version of the manuscript that addresses the points raised during the review process.

In the attached edited version of the revised manuscript, I have added final comments and changes to be made.

Responses to specific comments (line numbers with comments correspond to marked up copy with tracks on, line numbers for 2nd revision correspond to unmarked version without tracked changes).

1) Line 177. Source reefs do not consume mud crabs. This type of grammatical error is common in the manuscript, and should be corrected.

Correction acknowledged – document checked for similar errors

Line 157 2nd revision.

2) Line 193. These are contradictory. Just use complementary.

Correction accepted

Line 170 2nd revision.

3) Line 194. Define ‘sparse’. 

’sparse’ defined parenthetically as, “lacking other mediating factors”

Line 171 2nd revision

4) Line 203. This doesn’t make sense. Why would sand preclude responses?

Statement rephrased: “The substrate was withheld to limit responses of mud crabs solely to the presence of caged predators.”

Line 179 2nd revision

5) Line 240. What of the blocking factor in both experiments?

A statement addressing the blocking factor was was inserted : “Tests of the blocking effect based on separate ANOVAs for each of the two substrate types in which Treatment was a fixed factor and Block was a random factor showed the blocking factor to be unimportant (F=2.116, P = 0.116 oyster shell; F=1.701, P=0.189 limestone gravel).”

Line 215 2nd revision

6) Line 249. Use r2 and two-tailed test.

Statistical parameters edited accordingly, and n also provided.

Line 220 2nd revision

7) Line 250. Show data in a table either in text or as a supplement.

A reference to an S1 Table in the text points to a new table placed in the Supplementary information section which lists mean differences and paired t-tests of small vs. large oysters consumed across treatment levels.

Line 222 2nd revision

8) Line 260. Do you mean statistically significant?

Sentence rephrased: “Levene’s and F-tests for unequal error variances and heteroscedasticity were not significant for oyster mortality within the MPE ANOVA.”

Line 227 2nd revision

9) Line 263. Many units comprise a whole, not the other way around.

Sentence rephrased: “The ANOVA for the mean number of mud crabs consumed consisted of the four MPE treatments involving mud crabs.”

Line 230 2nd revision

10) Line 319. These statements are contradictory. Either you do or you don’t have an interaction between substrate and treatment.

Less contradictory and more nuanced phrasing used: “The mean number of oysters consumed differed for the MPE Treatment and Substrate factors, but not for their interaction (Table 1). Overall, more oysters were consumed in limestone gravel (15.6 ± 1.93 se) than in oyster shell (11.3 ± 1.27 se) across MPE treatments, except for the toadfish/mud crab (TF/MC) treatment for which five-fold more oysters were consumed in the oyster shell than in limestone gravel (t = 3.14; P1-t = 0.007) (Table 2; Figure 1).”

Line 269 2nd revision

P.S. The following justifies the use of post-hoc tests between factor levels of specific interest in the case of nonsignificant interactions, and is excerpted from:

Wei, J, Carroll, RJ, Harden, KK, Wu, G. Comparisons of treatment means when factors do not interact in

two-factorial studies. Amino Acids. 2012; 42: 2031–2035. doi:10.1007/s00726-011-0924-0.

“….. based on the general analysis of factor effects, no comparison among the treatment means is usually suggested when the two factors do not interact. However, this does not mean that comparisons among treatment means cannot or should not be made.…..a simple post hoc t test is sufficient to achieve the P value and also provides a greater power in statistical analysis. This new strategy can be used in the future studies involving multiple comparisons of treatment means when there is a nonsignificant interaction between two factors…..Comparison among treatment means when there is no interaction is meaningful for some specific situations.”

11) Line 358. Table 1. Define this in legend.

The corrected model term is now defined in the legend: “The corrected model term excludes variability explained by the intercept, thereby constituting an overall test of the dependent variables.”

Line 304 2nd revision

12) Line 363. Table 2. Add sample sizes so that standard errors can be computed. Better yet, add standard errors and sample sizes.

Two new columns added to Table 2 to show standard errors and sample sizes.

Line 308

13) Line 379. Table 3. What is the underline?

Underlined value replaced by bold value connoting P ≤ 0.05.

Line 324

14) Line 379. Table 3. Delete the superscripts.

Done.

Line 336

15) Line 419. This is confusing because Table 3 does not show substrate for all treatments. Please revise.

This paragraph has been extensively rewritten and expanded as two paragraphs to clarify the interpretation of the interaction effects in the mud crab PERMANOVA, and NCEs relative to mud crabs and oysters as prey.

Line 287

16) Line 466. Table 4. Define these response variables in the legend.

The following was appended to the Table 4 legend: “Predator present vs. predator absent values represent numbers of 15 mm oysters consumed during the experiments (out of 10 possible except for the mixed size treatment, which used 20 oysters).”

Line 376

17) Line 541. Citation?

Sentence restated and newly cited as [49]: “However, provided toadfish can grow large enough (i.e., > 300 mm TL) [23, 49] to pose a predatory risk to adult stone crabs in the field, crabs of the size used in our experiment may not have entirely outgrown the risk of predation by toadfish. Indeed, the large mouth width of the oyster toadfish equals 20% of its TL [49]. 

Line 439

18) Line 562. Define.

Explicitly defined the first time as “valve length”.

Line 457

19) Changes by academic editor made throughout the manuscript

All changes made by the Academic Editor within the revised manuscript were gratefully accepted.

---

## [Editor Report · Decision Letter 2]

13 Nov 2020

Destabilizing effects on a classic tri-trophic oyster-reef cascade

PONE-D-20-05873R2

Dear Dr. Rakocinski,

We’re pleased to inform you that your manuscript has been judged scientifically suitable for publication and will be formally accepted for publication once it meets all outstanding technical requirements.

Kind regards,

Judi Hewitt

Academic Editor

PLOS ONE
---

## [Editor Report · Acceptance letter]

4 Dec 2020

PONE-D-20-05873R2 

Destabilizing effects on a classic tri-trophic oyster-reef cascade 

Dear Dr. Rakocinski:

I'm pleased to inform you that your manuscript has been deemed suitable for publication in PLOS ONE. Congratulations! Your manuscript is now with our production department. 

Kind regards, 

on behalf of

Dr. Judi Hewitt 

Academic Editor

PLOS ONE